# COVID-Net Biochem: An Explainability-driven Framework to Building Machine Learning Models for Predicting Survival and Kidney Injury of COVID-19 Patients from Clinical and Biochemistry Data

**Hossein Aboutalebi**[*,1,3,6], **Maya Pavlova**[*,2,6], **Mohammad Javad Shafiee**[2,3,4,6],
**Adrian Florea**[5], **Andrew Hryniowski**[2,4], **Alexander Wong**[1,2,3,4,6]

[1]Cheriton School of Computer Science, University of Waterloo, Canada
[2]Department of Systems Design Engineering, University of Waterloo, Canada
[3]Waterloo Artificial Intelligence Institute, University of Waterloo, Canada
[4]DarwinAI Corp., Canada
[5]CIUSSS de l'Ouest-de-l'Île-de-Montréal
[6]{haboutal, mspavlova, mjshafiee, a28wong}@uwaterloo.ca
[*] Equal Contributions

## Abstract

A major challenge faced during the pandemic has been the prediction of survival and the risk for additional injuries in individual patients, which requires significant clinical expertise and additional resources to avoid further complications. In this study we propose COVID-Net Biochem, an explainability-driven framework for building machine learning models to predict patient survival and the chance of developing kidney injury during hospitalization from clinical and biochemistry data in a transparent and systematic manner. In the first "clinician-guided initial design" phase, we prepared a benchmark dataset of carefully selected clinical and biochemistry data based on clinician assessment, which were curated from a patient cohort of 1366 patients at Stony Brook University. A collection of different machine learning models with a diversity of gradient based boosting tree architectures and deep transformer architectures was designed and trained specifically for survival and kidney injury prediction based on the carefully selected clinical and biochemical markers. In the second "explainability-driven design refinement" phase, we harnessed explainability methods to not only gain a deeper understanding into the decision-making process of the individual models, but also identify the overall impact of the individual clinical and biochemical markers to identify potential biases. These explainability outcomes are further analyzed by a clinician with over eight years experience to gain a deeper understanding of clinical validity of decisions made. These explainability-driven insights gained alongside the associated clinical feedback are then leveraged to guide and revise the training policies and architectural design in an iterative manner to improve not just prediction performance but also improve clinical validity and trustworthiness of the final machine learning models. Using the proposed explainable-driven framework, we achieved 97.4% accuracy in survival prediction and 96.7% accuracy in predicting kidney injury complication, with the models made available in an open source manner. While not a production-ready solution, the ultimate goal of this study is to act as a catalyst for clinical scientists, machine learning researchers, as well as citizen scientists to develop innovative and trust-worthy clinical decision support solutions for helping clinicians around the world manage the continuing pandemic.[1]

---

[1]**The benchmark dataset created in this study and the link to the code is available here**

2022 Trustworthy and Socially Responsible Machine Learning (TSRML 2022) co-located with NeurIPS 2022.

# 1   Introduction

As stated by the recent studies (1; 2), COVID-19 infection can cause serious health complications like Acute kidney injury (AKI) which can be fatal in some patients. Understanding these complications and acting preemptively during the treatment can significantly increase the survival chance of a patient (3) suffering from COVID-19. Nevertheless, lack of resources makes taking swift actions even more difficult.

In this paper, a new machine learning model framework is proposed to predict a patient's survival chance and the chance of developing kidney injury during hospitalization from clinical and biochemistry data within a transparent and systematic manner. The proposed COVID-Net Biochem method is an explainability-driven framework for building machine learning model for the aforementioned tasks that can be extended to other healthcare domains. The explainability insight derived from the model decision-making process provides a framework to be able to audit model decisions. This capability can be used in tandem to gain new powerful insights of potential clinical and biochemical markers that are relevant to the prediction outcome. As such, the proposed method can assist physicians to make the diagnosis process more effective and efficient by providing supplementary outcome predictions based on a large collection of clinical and biochemical markers as well as highlighting key markers relevant to the task.

The resulting output from the framework includes a diverse collection of machine learning models including different gradient based boosting tree architectures and deep transformer architectures designed to specifically predict survival chance and kidney injury as well as their dominant clinical and biochemical markers leveraged throughout the decision making process.

In this work, we propose COVID-Net Biochem, an explainability-driven framework for building machine learning models for patient survival prediction and AKI (Acute Kidney Injury during hospitalization) prediction in a transparent and systematic manner. The proposed two-phase framework leverages both clinician assessment and deep insights extracted via a quantitative explainability strategy to not only gain a deeper understanding into the decision-making process of the machine learning models, and the impact of different clinical and biochemical markers on its decision-making process, but also enables the creation of high-performance, trustworthy, clinically-sound machine learning models by guiding architecture design and training policies based on these extracted clinical and explainability-driven insights in an iterative manner.

In particular, we take advantage of explainable AI in the architecture design and training process to ensure the final model decision process matches the clinical perspective. In this regard, there are other studies including (4), (5), and (6) that use explainiblity toolkit like GSInquire to validate the decision-making behaviour of their respective models in consultation with clinicians. In contrast, our proposed two-phase model building framework uses GSInquire to guide the refinement of both the data and model design through an iterative clinician-in-the-loop approach, effectively using explainability as a part of the model development process rather than as a final validation step. Notably, this closed-loop approach is applicable to any explainability algorithm, and as such, the main contribution is the model development framework rather than the use of GSInquire in particular. In this regard, for the first time, we have clearly outlined which phases of model building and dataset refinement the clinicians can get involved in, and how we can leverage their guidance in our model building to create more trustworthy machine learning models. This approach reduces the risk of bias and inconsistent model behavior, including using irrelevant or biased markers in the decision-making process.

**Generalizable Insights about Machine Learning in the Context of Healthcare**

A key generalizable insight we wish to surface in this work is the largely 'black box' nature of model design at the current state of machine learning in the context of healthcare, and the strategies for transparent design are not only critical but very beneficial for building reliable, clinically relevant models in a trustworthy manner for widespread adoption in healthcare. More specifically, while significant advances have been made in machine learning, particularly with the introduction of deep learning, much of the design methodologies leveraged in the field rely solely on a small set of performance metrics (e.g., accuracy, sensitivity, specificity, etc.) to evaluate and guide the design process of machine learning models. Such 'black box' design methodologies provide little insight into the decision-making process of the resulting machine learning models, and as such even the designers themselves have few means to guide their design decisions in a clear and transparency manner. This

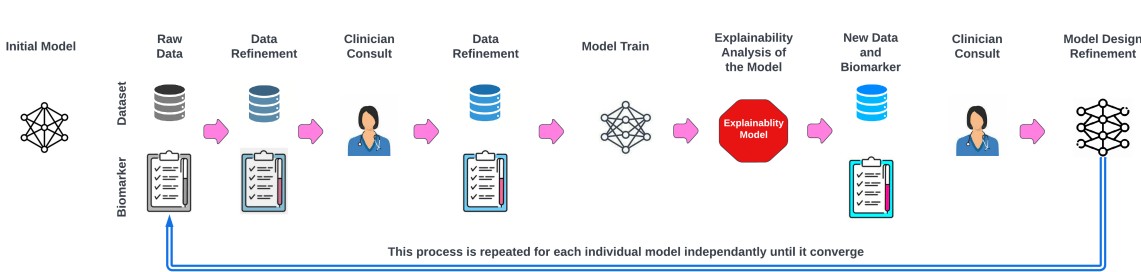

Figure 1: Overview of the proposed explainability-driven framework for building machine learning models for clinical decision support.

is particularly problematic given the mission-critical nature of clinical decision support in healthcare, and can lead to significant lack of trust and understanding by clinicians in machine learning-driven clinical decision support solutions. Furthermore, the lack of interpretability or understanding in the decision-making process during the design process and during clinical use creates significant accountability and governance issues, particularly if decisions and recommendations made by machine learning models result in negative patient impact in some cases.

Motivated to tackle the challenges associated with 'black box' model design for clinical decision support, in this work we propose an explainability-driven development framework for machine learning models that can be extended to multiple healthcare domains such as COVID-19 survival and acute kidney injury prediction. The framework provides a two phase approach in which a diverse set of machine learning models are designed and trained on a curated dataset and then validated using both an automatic explainability technique to identify key features as well as manual clinician validation of the proposed highlighted features. The second phase consists of leveraging the explainability-driven insights to revise the data and design of the models to ensure high detection performance from relevant clinical features.

## 1.1 Related Works

While using computer-aided diagnostics for screening and medical imaging of COVID-19 patients has been very popular(7; 8; 9; 10; 11; 12; 13; 14; 15), little work has been done in using machine learning models to assess the survival chance and prediction of development of acute kidney injury (AKI) among COVID-19 patients. Furthermore, most of the proposed algorithms so far lack interpretability which makes their real-world application and integration with clinicians questionable as their decision making process is not reliable.

One of the more relevant study to the proposed method is the approach introduced by Gladding *et al.* (16) which utilizes machine learning model to determine COVID-19 and other diseases from hematology data. Furthermore, Erdi *et al.* (17) proposed a novel deep learning architecture for detection of COVID-19 based on laboratory results. While these works focus on determining the COVID-19 positive cases, our work focuses on determining the survival chance and the chance of developing AKI during hospitalization based on biochemical data with the proposal of an end-to-end transparent model development framework that can be extended to other healthcare domains.

## 2 Explainability-driven framework for building machine learning models for clinical decision support

The proposed COVID-Net Biochem framework comprises of two main phases:

**Clinician-guided initial design phase:** The first phase starts with the preparation of a benchmark dataset from carefully selected clinical and biochemical markers based on clinical assessment curated for a patient cohort. While a plethora of clinical and biochemical markers may be collected for the patient cohort, only a selected number of markers are relevant for a given predictive task while others may be not only irrelevant but misleading for the machine learning model when leveraged. Therefore,

in this phase, we remove clinically irrelevant markers through consultations with clinicians who have the domain knowledge of the task. Next, a collection of different machine learning models with a diversity of gradient based boosting tree architectures and deep transformer architectures is designed and trained on the constructed benchmark dataset.

**Explainability-driven design refinement phase:** The second phase starts with the quantitative explainability validation of model performance and behaviour to gain a deeper understanding of the decision-making process, as well as gaining quantitative insights into the impact of clinical and biochemical markers on the decision-making process and the identification of the key markers influencing the decision-making process. In this paper, we leverage a quantitative explainability technique called GSInquire to conduct this evaluation. Next, we analyze and interpret the decision making process of the model through the identified relevant predictive markers and leverage the insights in an iterative design refinement manner to build progressively better and more clinically-relevant machine learning models. More specifically, if all of the clinical and biochemical markers identified by quantitative explainability are driving the decision-making process of a given model and these markers are verified to be clinically sound based on clinical assessment of the explainability results, the model is accepted as the final model; otherwise, we returns to the first phase where the irrelevant markers are discarded for that given model, and a new model architecture is trained and produced via hyperparameter optimization and again tested for phase 2. This iterative approach not only removes the influence of quantitatively and clinically irrelevant clinical and biochemical markers, but it also removes the markers that may dominate the decision-making process when they are insufficient for clinically sound decisions (e.g., the heart rate clinical marker may be clinically relevant but should not be solely leveraged for survival prediction from COVID-19 as a result of its general severity implication). This iterative process is continued until the model heavily leverages clinically sound clinical and biochemical markers to great effect and impact in its decision making process. Figure 1 provides an overview of the complete iterative design process in the proposed framework.

In this particular study, the clinician-guided initial design phase consists of constructing a new benchmark dataset of clinical and biochemistry data based on clinical feedback curated from a patient cohort of 1366 patients at Stony Brook University (18). The collection of models we designed and trained on the constructed benchmark dataset are based on the following architecture design patterns: i)TabNet (19), ii) TabTransformer (20), iii) FTTransformer (21), iv) XGBoost (22), v) LightGBM (23), and vi) CatBoost (24). In addition, for a baseline comparison, we added established models including Logistic regression and random forest in our results. TabNet focuses on employing sequential attention to score features for decision making and make the model more interpretable compared to previously proposed deep learning models for tabular datasets (19). TabTransformer and FTTransformer utilize a more recent transformer architecture designed to process Tabular datasets. In practice, transformer models have shown higher performance on most well-known datasets (20; 21; 25). The gradient boosting algorithms rely on creating and learning an ensemble of weak prediction models (decision trees) by minimizing an arbitrary differentiable loss function.

In the explainability-driven design refinement phase for this particular study, we conduct quantitative explainability validation of model performance and behaviour by leveraging GSInquire (26), a state-of-the-art explainability technique that has been shown to produce explanations that are significantly more reflective of the decision-making process when compared to other well-known explainability techniques in the literature. GSInquire enables the assignment of quantitative importance values to each clinical and biochemical marker representing its impact on the model prediction. Finally, clinical assessment of explainability-driven insights was conducted by a clinician with over eight years of experience.

**Note:** *Due to the lack of space the details of our data preparation and details of data refinement phase are included in the appendix.*

## 3  Experiment

In this section, we describe the experimental results and training procedure for the different machine learning models created using the proposed framework for the purpose of predicting COVID-19 patient survival and predicting the development of AKI (Acute Kidney Injury) in COVID-19 patients during hospitalization. As mentioned earlier, we designed six different machine learning models for the aforementioned prediction tasks using the following architecture design patterns: TabTransformer, TabNet, FTTransformer, XGBoost, LightGBM, and CatBoost. Our training procedure is guided

| Survival Prediction | | | | |
|---|---|---|---|---|
| Model | Accuracy | Precision | Recall | F1 Score |
| FTTransformer | 89.3% | 90.6% | 97.8% | 0.89 |
| TabTransformer | 96.7% | 97.8% | 98.3% | 0.96 |
| TabNet | 86.8% | 86.8% | 100.0% | 0.86 |
| XGBoost | **97.4%** | **97.5%** | 99.5% | **0.97** |
| LightGBM | 97.0% | 97.5% | 99.1% | 0.97 |
| CatBoost | 96.7% | 97.1% | 99.1% | 0.96 |
| Random Forest | 87.1% | 87.1% | **100.0%** | 0.87 |
| Logistic Regression | 87.5% | 88.8% | 97.8% | 0.87 |

Table 1: Accuracy, precision, recall, and F1 score of tested models for Survival Prediction

| Confusion Matrix XGBoost | | | | Confusion Matrix TabTransformer | | |
|---|---|---|---|---|---|---|
| Class | Deceased | Discharged | | Class | Deceased | Discharged |
| Deceased | 30 | 6 | | Deceased | 31 | 5 |
| Discharged | 1 | 236 | | Discharged | 4 | 233 |

Table 2: Confusion Matrix for CatBoost and TabTransformer for Survival Prediction

by not only accuracy, precision, and recall but also by identified explainability results. In the next section we provide explainability on the models decision process. **Most results are included in the Appendix.**

### 3.1 Survival prediction

We set the *last status* which had a binary value of deceased or discharged as our target in this task. For the training, as briefly discussed in the previous section, we constantly monitored the decision making process of the model using GSInquire to make sure the model is choosing the relevant set of features to make the prediction. Details of the setting for training is included in the Appendix.

The results for the models are depicted in Table 1. Also Table 2 shows the confusion matrix for CatBoost and TabTransformer. As it can be seen XGBoost had the best performance achieving accuracy of 97.4 % on the test set. Among deep learning models, TabTransformer had the best performance with accuracy of 96.7%. Also, both TabTransformer and XGBoost had above 96% results for recall and precision.

### 3.2 AKI prediction

We set the *Acute kidney injury during hospitalization* which had a binary value of True or False as our target in this task. The training procedure for this task was very similar to the survival task with hyperparameters almost the same. Except here we also removed the *last status* marker from our input to the models as it is a non-relevant clinical marker.

**Note:** Due to lack of space the results for AKI prediction is included in the Appendix.

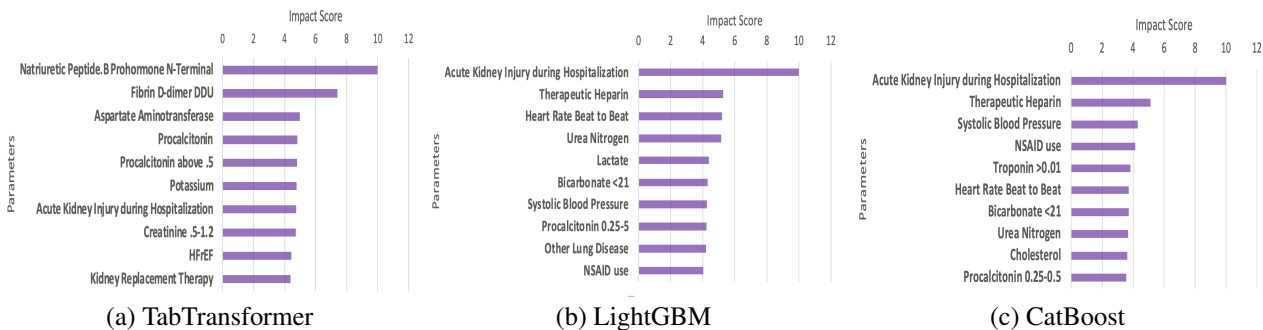

(a) TabTransformer    (b) LightGBM    (c) CatBoost

Figure 2: Top 10 markers identified through explainability-performance validation for TabTransformer, LightGBM, and CatBoost survival prediction models.

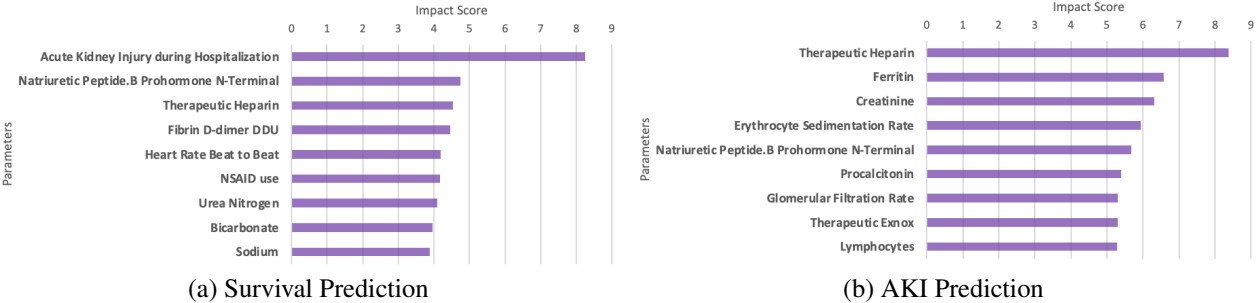

(a) Survival Prediction            (b) AKI Prediction

Figure 3: Top 10 most predictive clinical and biochemical markers averaged across models for COVID-19 patient survival and AKI prediction.

### 3.3 Explainability

As explained earlier, the trained models from phase one of the development framework were then audited via explainability-driven performance validation to gain insights into their decision-making process that will inform the design modifications in phase two of the process. We leveraged GSInquire (27) to provide quantitative explainability of input clinical and biochemical markers. More specifically, GSInquire provides impact scores for each marker based on their influence on the outcome prediction through an inquisitor $\mathcal{I}$ within a generator-inquisitor pair $\{\mathcal{G}, \mathcal{I}\}$. These actionable insights were then further validated by a clinician to ensure the clinical relevance and later employed by the framework to make design revisions to the models accordingly.

Figures 2 show the 10 most impactful clinical and biochemical markers relevant to COVID-19 survival prediction for the highest performing models of TabTransformer, LightGBM, and CatBoost. The results for AKI is included in the Appendix. Figure 3 provides a summary of the high impact markers across all models by averaging their impact scores and reporting the top 10 highest positive predictive parameters. For COVID-19 patient survival prediction, the marker indicating whether a patient has experienced acute kidney injury during hospitilization provides the highest impact to model predictions which is aligned with our clinician suggestion. In this regard, we observed in figure 4 (included in the Appendix), that there is a direct correlation between survival and acute kidney injury. Also, it is interesting to see that in figure 2 (included in the Appendix) while two gradient boosting tree has the same high impact marker (AKI), the tabtransformer is looking at two other markers B and Fibrin D Dimer which are also relevant for survival prediction. What we can see here is that as we change the type of learning model from gradient boosting tree to deep neural network, the model considers a different set of relevant clinical and biochemical markers. This also happens in 6 (included in the Appendix) for acute kidney injury prediction. While we can clearly see that both gradient boosting trees are considering the Theraputic Heparin and Creatinine to determine the chance of developing AKI, the TabTransformer is considering a different set of relevant markers such as Ferritin for decision making.

Finally, it is worth mentioning that our clinician found the figure 3 very interesting which represents the main markers used by all model on average. In this regard, most of the biomedical and clinical markers including Creatinine, Therapeutic Heparin used in this figure are considered as most relevant markers to determine survival rate of the patient and chance of AKI.

### 4 Discussion

In this work we presented an explainability-driven framework for building machine learning models which is able to build transparent models that only leverage clinically relevant markers for prediction. As a proof of concept, we applied this framework for predicting survival and kidney injury during hospitalization of COVID-19 patients such that only clinically relevant clinical and biochemical markers are leveraged in the decision-making process of the models and to ensure that the decisions made are clinically sound. In this regard, we provided a comprehensive examination of the constructed machine learning models' accuracy, recall, precision, F1 score, confusion matrix on the benchmark dataset. Furthermore, we interpreted the decision-making process of the models using quantitative explainability via GSInquire. Finally, we showed that the model uses acute kidney injury as the main factor to determine the survival chance of the COVID-19 patient, and leverages Creatinine biochemical markers as the main factor to determine the chance of developing kidney injury which is consistent with clinical interpretation.

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

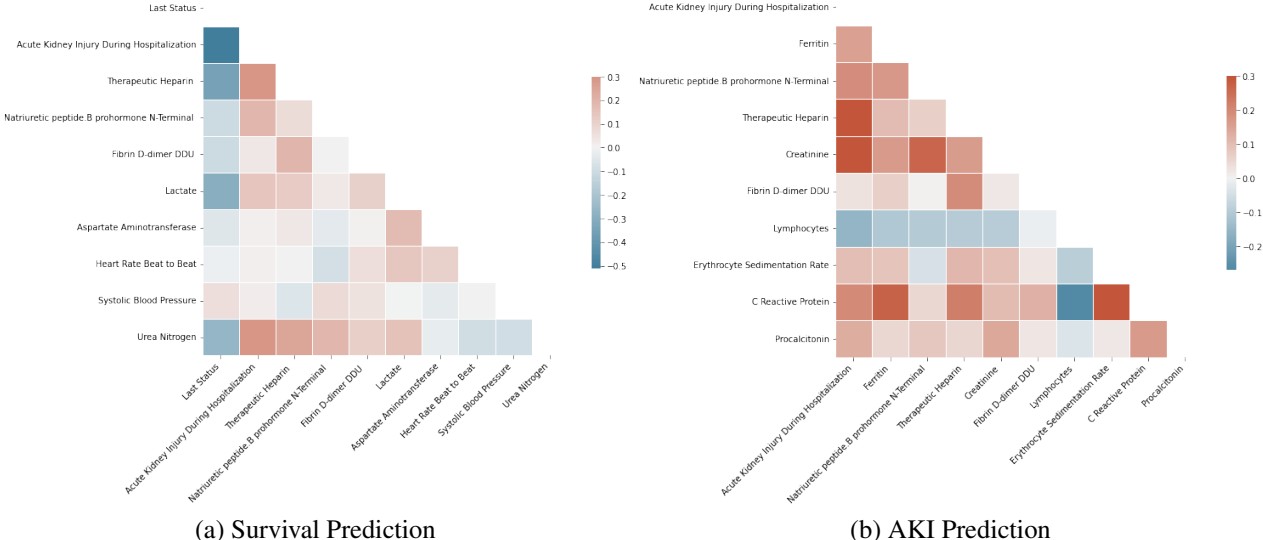

(a) Survival Prediction        (b) AKI Prediction

Figure 4: Pearson correlation coefficients between key identified clinical and biochemical markers for survival and AKI prediction.

# 5 Appendix

# 6 Data Preparation and Refinement

Due to the lack of space, the details of our clinician-guided design of dataset is included in the appendix.

In this section we provide a comprehensive overview of the data preparation process in constructing a benchmark dataset for COVID-19 patient survival and AKI prediction in the clinician-guided initial design phase of the proposed framework, as well as the clinical and biochemical marker selection process based on explainability-driven insights in the explainability-driven design refinement phase. The proposed dataset is built by carefully selecting clinical and biochemical markers based on clinical assessment from a patient cohort curated by Stony Brook University (18). More specifically, the clinical and biochemical markers were collected from a patient cohort of 1336 COVID-19 positive patients, and consists of both categorical and numerical markers. The clinical and biochemical markers include patient diagnosis information, laboratory test results, intubation status, oral temperature, symptoms at admission, as well as a set of derived biochemical markers from blood work. Table 3 demonstrates the numeric clinical and biochemical markers from the patient cohort and their associated dynamic ranges.

The categorical clinical markers consists of *"gender"*, *"last status"* (discharged or deceased), *"age"*, *"is icu"* (received icu or not), *"was ventilated"* (received ventilator or not), *"AKI during hospitalization"* (True or False), *"Type of Theraputic received"*, *"diarrha"*, *"vomiting symptom"*, *"nausea symptom"*, *"cough symptom"*, *"antibiotic received"* (True or False), *"other lung disease"*, *"Urine protein symptom"*, *"smoking status"*, and *"abdominal pain symptom"*.

**Target value**: In this study, the *"last status"* is used as a target value for the task of predicting the survival chance given the patient's symptoms and status. In addition, the *"AKI during hospitalization"* is identified as a target for the task of predicting the kidney injury development during hospitalization. Figure 5.a and figure 5.b demonstrates the distribution of these two target values in the patient cohort is highly unbalanced. **Missing Value and Input Transformation**: For replacement and modification, we found that using different type of input transformation does not substantially change the final result in our models. In this regard, we examined MinMax scaler, uniform transformer and normal distribution transformer (all available in sickit-learn preprocessing method (28)). None of them provided any better results.

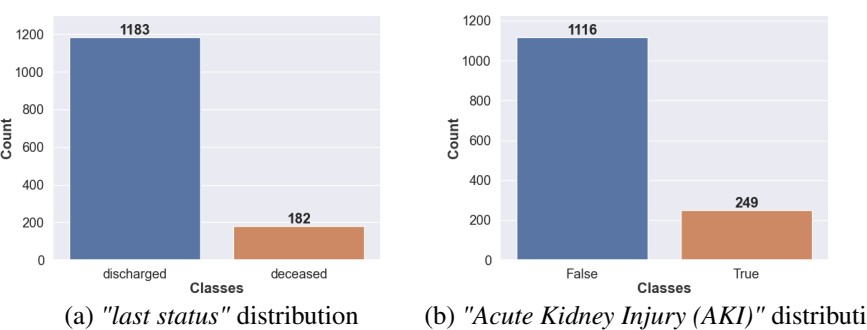

(a) *"last status"* distribution     (b) *"Acute Kidney Injury (AKI)"* distribution

Figure 5: Distribution of *"last status"*, *"AKI"*

On the other hand, the dataset had many missing values and to resolve the issue, for any marker that had more than 75% missing values in the dataset, the corresponding marker was removed in our study. For replacing missing value for our models, we found that both transformer models and gradient boosting tree models are resilient against missing value and replacing the missing value with a constant gives a competitive result. Particularly, we followed the same strategy introduced in TabTransformer (20) where the missing value is treated as an additional category.

## 6.1 Clinically guided marker selection

To create our benchmark dataset in the clinically guided initial design phase, we consulted with a clinician with over 8 years of experience and identified clinical markers that are clinically irrelevant and may result in biases being learnt by the machine learning models. More specifically, confounding factors such as *"heart rate", "invasive ventilation days"* were removed after consulting with the clinician as their impact on survival and AKI prediction were not directly clinically relevant.

## 6.2 Explainability-driven clinical and biochemical marker refinement

In the explainability-driven design refinement phase, we leverage quantitative explainability to analyse the decision-making processes of the individual trained models within the collection of initial model designs, and identified the most quantitatively important clinical and biochemical markers for each of the models using the GSInquire quantitative explainability method (27). After identifying the most quantitatively important markers to the decision-making processes of individual models, we presented these explainability results to the clinician to not only gain valuable clinical insights into the clinical soundness of the machine learning models but also to identify the non-relevant markers among these that the models rely on so that they will be excluded in the next round of model design refinement and training. As an example, after conducting explainability-driven assessment on the machine learning models with LightGBM and CatBoost architectures, we observed that the clinical marker *"Length of Stay"* had the highest quantitative impact on the decision-making process of said models for the AKI prediction (see Figure 7). After clinical consultation on this explainability-driven insight, we found out this clinical marker has little clinical value in determining the likelihood of AKI. As a result, in the next phase of model design and training, the *"Length of Stay"* marker was excluded. This process continued until only the relevant markers for our prediction tasks were utilized by the individual models. It is very important to note that explainability-driven assessment was conducted on each model independently, and as such the end result is that each model is uniquely tailored around what clinical and biochemical markers benefits them most from the set of possible markers. The final result is shown in figure 6 which shows the models are not dependant on irrelevant markers. More explanation will be provided in the Explainability section on these figures.

Finally, to better show the correlation between clinical and biochemical markers, Figure 4 shows the correlation of top ten markers for AKI (acute kidney injury during hospitalization) and last status target marker. As seen, for the target marker *"last status"*, AKI has the highest correlation. On the other hand, for the target marker AKI, *"Urine Protein", "Therapeutic Heparin","Fibrin D Dimer", "Creatinine"* and *"Glomerular Filtration"* have the highest correlation values. It is worth to note our

| Clinical/Biochemical Markers (Numeric) | Minimum Value | Maximum Value |
|---|---|---|
| Invasive Ventilation Days | 0 | 40 |
| Length of Stay | 1 | 96 |
| Oral Temperature | 34 | 39.8 |
| Oxygen saturation in Arterial blood by Pulse | 55 | 100 |
| Respiratory Rate | 11.0 | 95 |
| Heart Rate Beat by EKG | 6 | 245 |
| Systolic Blood Pressure | 55 | 222 |
| Mean Blood Pressure by Non Invasive | 40 | 168 |
| Neutrophils in Blood by Automated count | 0.36 | 100 |
| Lymphocytes in Blood by Automated count | 0.36 | 100 |
| Sodium [Moles/volume] in Serum or Plasma | 100 | 169 |
| Aspartate aminotransferase in Serum or Plasma | 8 | 2786 |
| Aspartate aminotransferase in Serum or Plasma | 8 | 2909 |
| Creatine kinase in Serum or Plasma | 11 | 6139 |
| Lactate in Serum or Plasma | 5 | 23.8 |
| Troponin T.cardiac in Serum or Plasma | 0.01 | 1.81 |
| Natriuretic peptide.B prohormone N-Terminal in Serum or Plasma | 5 | 267600 |
| Procalcitonin in Serum or Plasma Immunoassay | 0.02 | 193.5 |
| Fibrin D-dimer DDU in Platelet poor plasma | 150 | 63670 |
| Ferritin [Mass/volume] in Serum or Plasma | 5.3 | 16291 |
| Hemoglobin A1c in Blood | 4.2 | 17 |
| BMI Ratio | 11.95 | 92.8 |
| Potassium [Moles/volume] in Serum or Plasma | 2 | 7.7 |
| Chloride [Moles/volume] in Serum or Plasma | 60 | 134 |
| Bicarbonate [Moles/volume] in Serum | 6 | 43 |
| Glomerular filtration rate | 2 | 120 |
| Erythrocyte sedimentation rate | 5 | 145 |
| Cholesterol in LDL in Serum or Plasma | 12 | 399 |
| Cholesterol in VLDL [Mass/volume] in Serum | 8 | 79 |
| Triglyceride | 10 | 3524 |
| HDL | 10 | 98 |

Table 3: Example numerical clinical and biochemical markers collected from the patient cohort

trained models in the experiment section are actually utilizing these markers to do decision making as discussed in the Explainability section.

# 7 Experiment

## 7.1 Training

For the training, we used 20% of the dataset as the test set and another 5 % as validation set. For TabTransformer, TabNet, and FTTransformer we did a grid search to find the best hyperparameter. In this regard, we set the batch size to 256, the learning rate was set to 0.00015 and we run the models for 150 epochs with early stopping on the validation set. We used Adam optimizer for all tasks. The training procedure was done in parallel with getting explainibilty results for the model. In this regard, we discarded features *"heart rate", "length of stay", "invasive ventilation days"* as models tend to heavily rely on these less relevant factors for decision making.

For gradient boosting models XGBoost, CatBoost, and LightGBM, we used the default setting except for the learning rate. The learning rate of 0.35 gave us the highest accuracy for CatBoost. For XGBoost and LightGBM, we set the learning rate to 0.3 and 0.1 respectively.

| AKI Prediction | | | | |
|---|---|---|---|---|
| Model | Accuracy | Precision | Recall | F1 Score |
| FTTransformer | 83.8% | 59.3% | 38.0% | 0.83 |
| TabTransformer | 93.0% | 91.8% | 68.0% | 0.93 |
| TabNet | 78.0% | 33.3% | 20.0% | 0.78 |
| XGBoost | 96.7% | 95.5% | 86.0% | 0.96 |
| LightGBM | **96.7**% | 97.6% | **84.0**% | **0.96** |
| CatBoost | 92.6% | 89.4% | 68.0% | 0.92 |
| Random Forest | 83.5% | **1.0**% | 10.0% | 0.83 |
| Logistic Regression | 84.9% | 71.4% | 30.0% | 0.84 |

Table 4: Accuracy, precision, recall, and F1 score of tested models for AKI Prediction

| Confusion Matrix LightGBM | | | | Confusion Matrix TabTransformer | | |
|---|---|---|---|---|---|---|
| Class | False | True | | Class | False | True |
| False | 222 | 1 | | False | 220 | 3 |
| True | 8 | 42 | | True | 16 | 34 |

Table 5: Confusion Matrix for CatBoost and TabTransformer for AKI Prediction

### 7.1.1 AKI prediction

The results for the models are depicted in Table 4. Also Table 5 shows the confusion matrix for LightGBM and TabTransformer. As it can be seen LightGBM had the best performance achieving accuracy of 96.7 % on the test set. Also, among deep learning models, TabTransformer had the best performance with accuracy of 93.0%.

**The benchmark dataset created in this study and the link to the code is available here**

## 8   Explainability

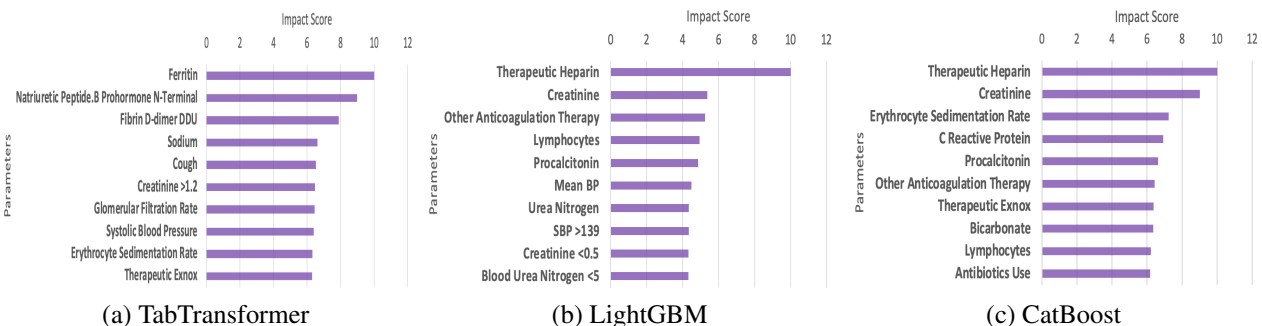

(a) TabTransformer        (b) LightGBM        (c) CatBoost

Figure 6: Top 10 markers identified through explainability-performance validation for TabTransformer, LightGBM, and CatBoost models leveraged for AKI prediction.

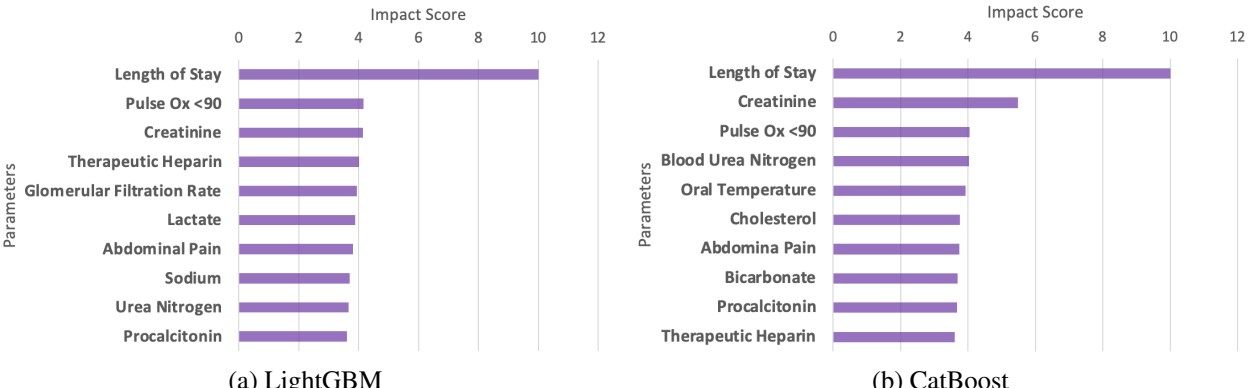

(a) LightGBM  (b) CatBoost

Figure 7: Top 10 clinical and biochemical markers identified through explainability-performance validation for LightGBM and CatBoost models for AKI prediction with the inclusion of the length of stay parameter in available patient data.

