# OpenReview forum: "COVID-Net Biochem: An Explainability-driven Framework to Building Machine Learning Models for Predicting Survival and Kidney Injury of COVID-19 Patients from Clinical and Biochemistry Data"
_NeurIPS.cc/2022/Workshop/TSRML — TSRML2022_

### Official Review · Reviewer_mMLd · 2022-10-15
**A good dataset for COVID-19 survival and kidney injury prediction**

**Overall Recommendation:** The data is promising and the framewo…
**Overall Rating:** 6

**Summary:**

This paper built a two-phase framework to give explainable COVID-19 survival and the risk of additional injury predictions. A dataset covering the cohort of 1366  patients has been collected and used for validation. As shown in this paper, the proposed framework yielded an outstanding performance, demonstrating the excellent promise for automatic and affordable pandemic management.

**Strengths:**

1. This paper collected a curated benchmark dataset for COVID-19-related research. Abadunt models have been implemented and compared to be baselines. In this regard, this paper could have a broader impact in both the clinical and machine learning fields.
2.  Explanibility is important yet usually ignored when building machine learning for health. The designed framework integrates explainability analysis, which provides powerful insights.




**Weaknesses:**

1. The writing of this paper can be improved. The introduction is too long with an unclear structure. It is understandable not to give details about the data due to the page limitation, but it is necessary to introduce what type of information is included, otherwise, it is hard to follow the experiments (I have to look at the appendix, although ideally, it is not recommended).
2. In Tables 1&2, many algorithms have been implemented. However, there is not much discussion shielding the light on their pros and cons.  A reasonability analysis is needed for future work.
3. For the classification, robustness results like the confidence interval or the performance variation are desired but missed.
4. Some related works need to be discussed, e.g.,
Näppi, Janne J., et al. "U-survival for prognostic prediction of disease progression and mortality of patients with COVID-19." Scientific reports 11.1 (2021): 1-11.
Nemati, Mohammadreza, Jamal Ansary, and Nazafarin Nemati. "Machine-learning approaches in COVID-19 survival analysis and discharge-time likelihood prediction using clinical data." Patterns 1.5 (2020): 100074.
Heller, Raban Arved, et al. "Prediction of survival odds in COVID-19 by zinc, age and selenoprotein P as composite biomarker." Redox biology 38 (2021): 101764.






**Review Confidence:**

4: The reviewer is confident but not absolutely certain that the evaluation is correct

---

### Official Review · Reviewer_AgLJ · 2022-10-21
**Interesting topic, poor writing, slightly lack of method validation**

**Overall Rating:** 6

**Summary:**

This paper proposes explainability-driven machine learning models for survival and kidney injury prediction of COVID-19 patient using clinical and biochemistry data.  Specifically, the data model includes two phase: the first phase was trained using machine learning to achieve the prediction task as well as selecting the most relevant biomarkers for the task; and the second phase involves the validation of these relevant biomarkers form an clinician. The model is trained iteratively with both the objective metrics such as accuracy as well as the evaluation from the clinician to gain deep understanding and increase the explainablity of the model.

**Strengths:**

The data was carefully selected, and it has great value. Further, including the clinician-in-the-loop mechanism for health-related tasks is sound and help advance the real deployment of the models in the clinical settings. The authors have also conducted extensive experiments using different modelling approaches for the task.

**Weaknesses:**

•	The paper is not well written, and a bit hard to follow.

•	In terms of selecting the most relevant features for the task, there are also many other approaches such as linear regression, feature importance or some specific feature selection techniques such as Feature subset selection (FSS). How does the proposed model compared to these simpler approaches to address the similar problem?


General:
•	Introduction is repetitive, e.g., the beginning of paragraph 2 and 4.

•	Format needs improvement, e.g., intro has one subsection without numbering and one with numbering, table formats can be improved, etc.



**Overall Recommendation:**

The paper addresses an important problem and proposes an explainable framework. Despite the relatively poor writing quality, the content and approach are interesting and promising.

**Review Confidence:**

4: The reviewer is confident but not absolutely certain that the evaluation is correct

---

### Decision · Program_Chairs · 2022-10-23

Accept